# A Phylogenetic and Morphological Evolution Study of *Ribes* L. in China Using RAD-Seq

**DOI:** 10.3390/plants12040829

**Published:** 2023-02-13

**Authors:** Baoshan Zhang, Ziyang Yu, Zhichao Xu, Baojiang Zheng

**Affiliations:** 1College of Life Science, Northeast Forestry University, Harbin 150040, China; 2Key Laboratory of Sustainable Forest Management and Environmental Microorganism Engineering of Heilongjiang Province, Northeast Forestry University, Harbin 150040, China; 3Northeast Asia Biodiversity Research Center, Harbin 150040, China

**Keywords:** *Ribes* L., phylogenetic, plant morphology, evolution, RAD-seq, China

## Abstract

*Ribes* L. belongs to the Grossulariaceae family and has important edible, medicinal, ornamental, and landscaping values. Taxonomic classification within this genus is difficult due to its large variety of species, wide distribution, large morphological variations, and presence of two complex taxonomic groups with bisexual or unisexual flowers. Our study aims to clarify the phylogenetic relationships of *Ribes* L. taxa in China, and further, to provide a reference for a revised global classification of it. The phylogenetic analysis of 52 *Ribes* L. samples from 30 species was constructed based on restriction site-associated DNA sequencing and single nucleotide polymorphisms. Afterward, two important taxonomic characters were selected for ancestral state reconstruction over the molecular phylogeny. The results showed that the 52 samples could be divided into six branches, i.e., six subgenera, which caused some controversy regarding the morphological classification of *Ribes* L. in China. The molecular phylogeny supported the separation of subg. *Coreosma* from subg. *Ribesia* and subg. *Hemibotrya* from subg. *Berisia* and validated the rationale for recognizing subg. *Grossularia* as an independent subgenus, the rationality of which was further verified by the reconstruction of ancestor traits. Gene flow among *Ribes* L. was identified and further confirmed our results.

## 1. Introduction

There are approximately 200 species of *Ribes* L. in the world, and they are primarily distributed in East Asia, North America, and the Andes of South America [1]. China is a distribution centre of *Ribes* L., with approximately 59 species and 30 varieties [2], including the main taxa in phylogenetic development. The economic value of this genus is high enough, as its fruits can be eaten raw or used to produce fruit wine, beverages, candies, and jams since they are rich in various vitamins, sugars, and organic acids [3,4].

Meanwhile, the roots and seeds of some species can also be used for medicinal purposes, while some species have high ornamental value due to their bright flowers and attractive fruits [5]. Therefore, clarifying the phylogenetic relationship within *Ribes* L. would be beneficial for the protection and rational use of the plant resources in this genus.

Linnaeus initially established the genus *Ribes* L. into eight species in 1753. Throughout the years, the classification of taxa within this genus remains a hot topic of debate [6,7,8,9,10]. Janczewski [6] classified *Ribes* L. gathered across the world into six subgenera: *Grossularioides*, *Parilla*, *Berisia*, *Coreosma*, *Ribesia*, and *Grossularia*, according to the characteristics of the flower, including sexuality, the texture of the bud scales as well as the presence or absence of a pedicel, pedicel joints, and thorns on the branches. Rehder [11] categorized this genus into four subgenera and 15 sections; in his classification, subg. *Coreosma* and subg. *Parilla* were placed in subg. *Ribesia* and subg. *Berisia*, respectively, which were different from the classifications by Janczewski. Berger [8] divided *Ribes* L. into eight subgenera: *Grossularioides*, *Parilla*, *Berisia*, *Coreosma*, *Ribesia*, *Calobotrya*, *Heritiera, and Symphocalyx* and *Grossularia* into four subgenera: *Grossularia*, *Hesperia*, *Lobbia*, and *Robsonia.* Weigend [7] classified *Ribes* L. into seven subgenera: *Ribes*, *Coreosma*, *Calobotrya*, *Symphocalyx*, *Grossularioides*, *Grossularia*, and *Parilla*. Lu [2] accepted the four subgenera classified by Rehder through an analysis of the morphological characteristics and the distribution of *Ribes* L. in China, although disagreed with the taxonomic ranks assigned by Rehder.

The taxa in *Ribes* L. are difficult to classify with morphological methods due to the high similarity in morphological characteristics among the species of this genus, large morphological variation, and the presence of two complex taxonomic groups with bisexual or unisexual flowers [12]. To date, some molecular markers have been developed to study the phylogenetic relationship of *Ribes* L. using technologies such as random amplified polymorphic DNA (RAPD) [13,14], 5S rDNA non-transcribed spacer (NTS) [15], internal transcribed spacer (ITS) [1], and chloroplast DNA loci [16]. In recent years, an increasing number of studies on the molecular systematics of *Ribes* L. in China were excavated that deeply promote its detailed classification [17,18]. However, the deep understanding of the inference of phylogenetic relationships or the taxonomic classification of *Ribes* L. is still obscure, for a lack of experimental materials and the limited number of polymorphic loci generated by the existing molecular markers. Therefore, it is necessary to develop more genomic data resources, including high-throughput genomic markers, to promote molecular systematic studies of *Ribes* L. at the genomic level.

In recent years, second-generation sequencing technology has promoted genome-wide biological research, in which high-throughput sequencing technology has been continuously applied in animal and plant genomics research [19], such as restriction site-associated DNA sequencing (RAD-seq), which is used in our study [20]. Briefly, RAD-seq is a technology used to construct the RAD sequencing library using a certain size of DNA fragments obtained through the digestion of the genome by restriction enzymes, then, the RAD markers produced after digestion are subjected to high-throughput sequencing [21]. Compared with others, the experimental process of RAD-seq is relatively simple and can be widely sampled throughout the genome without relying on the information of the reference genome [22]. Moreover, RAD-seq can obtain thousands of single nucleotide polymorphism (SNP) loci, making it an economical and effective method to identify large-scale SNPs and reduce genomic complexity and subtype genes [23]. Therefore, the simplified genome sequencing technology of RAD-seq has been reported to solve a series of genomics problems in a variety of organisms and is widely used in the study of genetic evolution, and phylogeny, alongside the species definition of wild populations and non-traditional research species [24,25,26].

Herein, we used RAD-seq to identify a large number of SNP loci from 52 samples of *Ribes* L., then, constructed phylogenetic relationships and analyzed the gene flow and genetic diversity among *Ribes* L. species. Furthermore, the evolution of key morphological characters was also traced to verify our classification of *Ribes* L. This study will provide important information on the classification of different species in this genus, enabling us to develop and utilize its germplasm resources rationally.

## 2. Results

### 2.1. RAD-Tag Sequencing and SNP Discovering

In this study, the raw data from 52 *Ribes* L. samples of 26 species and 4 varieties were filtered, with a total of 86.98 G of clean data obtained. We acquired 590.36 million clean reads by using Illumina HiSeq4000, after removing the low-quality reads (Q score < 20), and ambiguous reads with incorrect barcodes (Table 1). The sequencing quality scores of 20 (Q_20_), which represent an error rate of 1 in 100, with a corresponding call accuracy of 99%, with all samples at more than 97.72%, indicating that the sequencing quality was good. Of these high-quality reads, the highest reads (22.03 million reads) were detected in *Ribes nigrum* (D5), and the lowest reads (0.7 million reads) were found in *R. pseudofasciculatum* (B5), with an average read number of 11.35 million per accession.

A total of 2,451,454 high-quality SNPs were obtained, among them, 2,335,324 and 116,130 SNPs were homozygous and heterozygous, respectively (Table 1). The average number of detected SNPs was 47,713 per accession. The highest number of SNPs (168,263) was detected in *Ribes stenocarpum* (C5), while the lowest number of SNPs (5048) was detected in *R. pseudofasciculatum* (B5).

### 2.2. Phylogenetic Trees and Morphological Characteristics

Two phylogenetic trees of *Ribes* L. were constructed based on neighbor-joining and maximum likelihood analyses, while almost the same results were obtained. It showed that the 52 samples were clearly divided into two major branches and six subgenera, which correlated with the morphological characteristics (Figure 1). The first branch contained subg. *Ribesia*, *Ribes griffithii* Hook. f. et Thoms., subg. *Coreosma*, and subg. *Grossularia*, and the second branch contained subg. *Berisia*, and subg. *Hemibotrya*. A total of 23 samples representing nine species were clustered into subg. *Ribesia*, including *R. moupinense*, *R. setchuense*, *R. altissimum*, *R. himalense*, *R. longiracemosum*, *R. mandshuricum*, *R. palczewskii*, *R. triste*, and *R. atropurpureum*. The common morphological characteristics of subg. *Ribesia* species were having a bisexual flower and a raceme. Previous studies have also classified *R. griffithii* into subg. *Ribesia* [2,7]; however, our phylogeny showed that *R. griffithii* formed a monophyletic clade.

The subg. *Coreosma* included *Ribes nigrum*, *R. procumbens, and R. fragrans*, representing a total of three species (five samples), with common morphological characteristics, such as having bisexual flowers, racemes, glands on the abaxial leaf epidermis, and purple or black fruits. The subg. *Grossularia* contained *R. aciculare*, *R. burejense*, and *R. stenocarpum*, representing a total of three species (five samples), with common morphological characteristics, such as having thorny branches and inverted sepals.

A total of 15 samples representing 12 species were clustered into subg. Berisia, including *Ribes kialanum*, *R. tenue*, *R. takare* var. *desmocarpum*, *R. pseudofasciculatum*, *R. hunanense*, *R. davidii*, *R. laurifolium*, *R. laurifolium* var. *yunnanense*, *R. heterotrichum*, *R. giraldii* var. *polyanthum*, *R. alpinum*, *and R. komarovii*, with common morphological characteristics such as unisexual flowers and racemes. Among them, the samples of sect. Davidia were clustered on the same subbranch, with common morphological characteristics, such as being evergreen, having unisexual flowers, and having drooping inflorescences. This result was consistent with the clustering of sect. Davidia species into a section of subg. Berisia by Lu [2]. The other branch contained two species of the subg. Hemibotrya, namely *R. fasciculatum* and *R. fasciculatum* var. *chinense*, with common morphological characteristics such as unisexual flowers and umbels.

### 2.3. Principal Component Analysis of Ribes L.

PCA, using the first and second eigenvectors, identified six groups, i.e., subg. *Hemibotrya*, subg. *Berisia*, subg. *Grossularia*, subg. *Coreosma*, *Ribes griffithii* and subg. *Ribesia*, which were consistent with the phylogenetic clades (Figure 2A). Each group was represented by a different color. The 52 *Ribes* L. samples were clearly divided into two major categories. The PCA plot illustrated that the subg. *Grossularia* were more disperse than the other groups (Figure 2A).

Each point in the figure represents an individual species. The individual species in each group are well clustered together, showing high consistency. Group 1 consisted of two species of the subg. *Hemibotrya*, i.e., *Ribes fasciculatum* var. *chinense* and *R. fasciculatum*. Group 2 consisted of 12 species of the subg. *Berisia*. Group 3 was made up of three species of the subg. *Grossularia*, namely *R. aciculare, R. stenocarpum,* and *R. burejense*. Group 4 consisted of three species of subg. *Coreosma*, with a total of five samples. Group 5 contained *R. griffithii*. Group 6 was made up of 10 species of the subg. *Ribesia*. The subg. *Hemibotrya* and subg. *Berisia* clustered closely together (Figure 2B). The PCA analysis indicated that the genetic relationship between subg. *Hemibotrya* and subg. *Berisia* was relatively close. *R. griffithii*, subg. *Coreosma*, and subg. *Ribesia* formed three separate groups. The results showed that *R. griffithii* and subg. *Ribesia* were relatively separate.

### 2.4. Population Genetic Structure of Ribes L.

Each group is represented by a different color. The 52 *Ribes* L. samples were clearly divided into two major categories. The PCA plot illuminated that subg. *Grossularia* was more dispersed than the other groups (Figure 2A). Through population structure analysis, we could clarify how the *Ribes* L. species were clustered and, thus, we could understand the individual ancestries of the different species (Figure 3). For *K* = 2, the 52 *Ribes* L. samples in the figure were clustered into two categories, and the taxa with a blue background included all the samples from subg. *Ribesia*. Specifically, *Ribes griffithii* displayed an admixture of subg. *Berisia*, while subg. *Hemibotrya*. *R. griffithii* had a 5% brown background hybridization, supporting the classification of *R. griffithii* as a monophyletic clade. The taxa with brown backgrounds were subg. *Coreosma*, subg. *Grossularia*, subg. *Berisia*, and subg. *Hemibotrya*.

When *K* = 3, subg. *Berisia* and subg. *Hemibotrya* showed different genetic backgrounds from subg. *Coreosma* and subg. *Grossularia*, and all the samples were clearly clustered into four groups. At *K* = 4, hybridization occurred in some samples. For *K* = 5, *Ribes moupinense* and *R. setchuense* showed different pedigree compositions from those of other species in subg. *Ribesia*. When the *K* value increased, subg. *Coreosma* and subg. *Berisia* were separated, showing an independent genetic background. The optimal value of the population genetic structure analysis was at *K* = 7, and the ancestral compositions of *R. himalense*, *R. mandshuricum, and R. longiracemosum* differed from those of *R. altissimum* and *R. atropurpureum*, indicating that the former three species had closer genetic relationships. The results showed that subg. *Ribesia* has more interspecific hybridization. For *K* = 7, subg. *Hemibotrya* was separated from subg. *Berisia* with the hybridization in contrast to the observations at *K* = 4. At different *K* values, subg. *Berisia* contained a single color, confirming its independent genetic background.

### 2.5. Ancestral Character Reconstruction

Two key morphological characters of *Ribes* L. were selected to reconstruct their ancestral characters. As for the glands, we found that the lower leaf surface, calyx, ovary, and fruit of the three species of subg. *Coreosma* were densely covered with yellow glands, while other subgenus had no glands on the surface of plant bodies. The ancestral state reconstructions (Figure 4A) identified glandless as the most probable ancestral state for each subgenus of *Ribes* L. (subg. *Ribesia*: *p* = 0.99; *Ribes griffithii*: *p* = 0.99; subg. *Grossularia*: *p* = 0.99; subg. *Berisia*: *p* = 0.99; subg. *Hemibotrya*: *p* = 0.99). However, the gland state was reconstructed as the present state in the clade of subg. *Coreosma* (*p* = 0.99), which was clearly distinguished from the other subgenera. Specifically, glandular morphology of subg. *Coreosma* transitioned from glandless to gland, while there was no reversion to a glandless state.

The morphological characteristics of subg. *Hemibotrya* were significantly different from those of subg. *Berisia*. In terms of inflorescence type (Figure 4B), raceme was reconstructed as the ancestral state of *Ribes* L. (*p* = 0.99). Specifically, the inflorescences of three plants in the clade of subg. *Grossularia* changed from raceme (ancestral state) to short raceme independently (*p* = 0.98). However, the inflorescences of two plants of subg. *Hemibotrya* changed from raceme to umbel differently (*p* = 0.99). Neither subg. *Hemibotrya* nor subg. *Grossularia* reverted into the raceme. This indicates that each genus has had a high degree of independence in their own evolutionary history.

## 3. Discussion

### 3.1. Phylogenetic Relationships and Taxonomy of Ribes L.

The subgenus classification of *Ribes* L. has been controversial. Janczewski [6] classified *Ribes* L. into six subgenera based on the comprehensive analysis of some global morphological traits of *Ribes* L. Indeed, Berger [8] classified Grossulariaceae into two genera, namely, *Ribes* L. and *Grossularia*, and further divided *Ribes* L. into eight subgenera and *Grossularia* into four subgenera. Rehder [11] merged subg. *Coreosma* into subg. *Ribesia* and subg. *Hemibotrya* into subg. *Berisia*, thereby dividing *Ribes* L. into four subgenera. Weigend [7] classified *Ribes* L. into seven subgenera based on the morphological and micromorphological characteristics of this genus. Lu [2] clustered *Ribes* L. in China into four subgenera according to plant morphological characteristics. Huang [27] also divided *Ribes* L. into four subgenera based on an analysis of the pollen traits for 21 taxa of the genus. In contrast to previous studies, we divided 30 representative *Ribes* L. species in China into six branches based on phylogenetic trees by RAD-seq (Figure 1). In summary, for the relationships between these species, all three analyses, shown above, demonstrate similar patterns (Figure 2, Figure 3, and Figure 5). *Ribes* L. was seen to have two large complex taxonomic groups with unisexual or bisexual flowers. In Figure 1, the species with unisexual flowers are clustered into two branches, i.e., subg. *Hemibotrya* and subg. *Berisia*, which differ from the conclusions of Lu [2] and Huang [27] since these are based on morphology or micromorphology. We support the viewpoints of Berger [8] and Soltis [1]. We found that three samples of subg. *Hemibotrya* showed a hybrid genetic background from subg. *Berisia* and subg. *Coreosma*, although subg. *Hemibotrya* was not clustered with subg. *Berisia* on the phylogenetic trees or in the PCA diagram (Figure 2 and Figure 5). *Ribes fasciculatum* var. *chinense* and *R. fasciculatum* exhibited a different genetic background from other species of subg. *Berisia*. When *K* = 5, species of subg. *Hemibotrya* began to show hybridization. Therefore, we support the classification of subg. *Hemibotrya* as independent subgenus separate from subg. *Berisia*.

Huang [27] and Lu [2] allocated the *Ser. Nigra* species, namely, *Ribes procumbens*, *R. nigrum*, and *R. fragrans* in subg. *Ribesia*, thereby clustering *Ribes* L. into four subgenera. Berger [8] and Messinger [13] support placing these three species in subg. *Coreosma*, an independent subgenus. In addition, we found these three species of *Ribes* L. were not clustered in the same branch as other subg. *Ribesia* species (Figure 1 and Figure 2). The genetic structure analysis showed that with the increase in the *K* value, these three *Ribes* L. species showed an independent genetic background distinguished from that of other subg. *Ribesia* species. Based on the results of the phylogenetic analysis, we support placing these three *Ribes* L. species in subg. *Coreosma* instead of subg. *Ribesia*. Lu [2] classified *R. griffithii* as subg. *Ribesia*. We found it differed to subg. *Ribesia* in terms of some important morphological characteristics, including large changes in bract morphology, an oval, or lingual line to lanceolate, bracteoles, nectaries in flowers. Furthermore, we found that *R. griffithii* was formed by the hybridization of two ancestral populations (Figure 3), which was significantly different from subg. *Ribesia*. The taxonomic status of *R. griffithii* in *Ribes* L. needs to be further studied by expanding the sampling range.

Previous studies have classified gooseberries into the genus *Grossularia* or subg. *Grossularia* due to their unique morphological characteristics [7,8]. Our phylogenetic trees showed that *Ribes burejense*, *R. stenocarpum*, and *R. aciculare in subg. Grossularia* have relatively close phylogenetic relationships with subg. *Ribesia*, *R. griffithii*, and subg. *Coreosma*. In the PCA diagram, the five gooseberry samples (C1–C5) exhibited a scattered distribution, with distances far from the samples of other subgenera. The analysis of the population genetic structure showed that the genetic backgrounds of the five samples became independent when *K* = 5. This result confirmed a very close phylogenetic position and the relationship between them and each subgenus. Based on the above considerations, we are in favor of placing gooseberry plants into subg. *Grossularia* rather than considering them as an independent genus of *Grossularia*.

### 3.2. Gene Exchange Analysis of Ribes L.

China is a primary distribution center of *Ribes* L. in East Asia, with approximately 59 species and 30 varieties [2]. Its species abundance, with its many endemic species and large-scale overlap of the distribution areas of some species, has led to interspecific gene exchange in plants of the same genus [28]. We found that hybridization in subg. *Ribesia* is more complex (Figure 3 and Figure 5). When the *K* value increases, *Ribes altissimum* and *R. atropurpureum* had genetic backgrounds that differed from that of other subg *Ribesia* species and were not hybridized. For *K* = 5, *R. triste* and *R. palczewskii* exhibited a closer phylogenetic relationship, indicating the same ancestral origin. *R. mandshuricum* occupied three colors: blue, brown, and cyan, indicating a complex pedigree and more gene exchange with other species of *Ribes* L.

Regarding the *Ribesia* species, in the previous classification results, *Ribes griffithii* was placed in subg. *Ribesia* [2,7]. In Figure 1, *R. griffithii* and other subg. *Ribesia* plants were divided into two branches. Population genetic structure showed that hybridization consistently occurred in *R. griffithii* (Figure 3 and Figure 5). When *K* = 7, *R. griffithii* exhibited a histogram composed of two colors, cyan and yellow, and the genetic background of both subg. *Berisia* (20%) and subg. *Ribesia* (80%). Therefore, the sample size must be expanded for further analysis of the taxonomic position of *R. griffithii*.

The histograms corresponding to *Ribes fasciculatum* and *R. fasciculatum* var. *chinense* in subg. *Hemibotrya* were composed of two colors, green and yellow, indicating that they likely resulted from the hybridization of the two ancestors, subg. *Berisia* (80%) and subg. *Coreosma* (20%). This finding also indicated that subg. *Hemibotrya* should not be merged into subg. *Ribesia*, confirming the result of the phylogenetic tree (Figure 1). Overall, the pedigrees of subg. *Grossularia*, subg. *Coreosma*, and subg. *Berisia* were relatively pure. The subg. *Hemibotrya* might be formed by the hybridization of subg. *Berisia* and subg. *Coreosma*. The subg. *Ribesia* presented complex genetic backgrounds and frequent interspecific hybridizations. The three parallel analyses (phylogenetic, principal component, and genetic structure) provided comprehensive molecular evidence regarding the six groups (subg. *Ribesia*, *R. griffithii*, subg. *Coreosma*, subg. *Grossularia*, subg. *Berisia*, and subg. *Hemibotrya*).

### 3.3. Revision of subg. Hemibotrya

The views on the classification of *Ribes fasciculatum* and *R. fasciculatum* var. *chinense* seemed to differ widely among researchers [2,11]. Initially, they were divided into subg. *Berisia*, yet were also clearly distinguished from the other species of subg. *Berisia* because of their morphological characteristics of umbels. Moreover, some researchers had suggested that these two Ribes should be subg. *Parilla*, separately [6,29]. However, in our study we found that *R. fasciculatum* and *R. fasciculatum* var. *chinense* did not cluster with other species of subg. *Berisia* (Figure 1 and Figure 2). The genetic structure analysis also showed the interspecific hybridization between these two species, which was different from other species of subg. *Berisia* (Figure 3 and Figure 5). Furthermore, Figure 4 also shows that the inflorescence type of these two plants had changed from racemes to umbels (*p* = 0.99).

In addition, Weigend thought that dividing *Ribes fasciculatum* distributed in East Asia into the subg. *Parilla* would cause a mixture of species in this subgenus; therefore, suggested that *R. fasciculatum* should be separated from subg. *Parilla* [15]. Soltis analyzed the ITS sequences of 66 species of plants in 12 subgenera of *Ribes* L. and found that *R. fasciculatum* did not cluster with other species of subg. *Berisia* [1], which was consistent with our results. Therefore, Soltis supported the independence of *R. fasciculatum* as a sister species to the other species of subg. *Berisia*. Combined with the results of this study, we are in favor of the classification of *R. fasciculatum* and *R. fasciculatum* var. *chinense* as a separate subgenus, rather than classifying them into subg. *Berisia* or subg. *Parilla*.

## 4. Materials and Methods

### 4.1. Plant Materials and DNA Isolation

For this study, we collected 52 representative samples of 30 *Ribes* L. species from 15 provinces in China (Figure 6 and Table 2). It is worth mentioning that although some species are remote and difficult to collect, the 30 species we collected and analyzed depict nearly all the representative species of each branch in China. The plant names in this study follow the nomenclature of Flora of China [30]. Genomic DNA was extracted using the cetyltrimethylammonium bromide (CTAB) method [31]. Following quality assessment, the DNA concentration was adjusted to 100 ng/μL for RAD-seq library preparation.

### 4.2. RAD Library Preparation and Sequencing

A reduced representation restriction-associated DNA (RAD) sequencing method was used for library construction following the protocol previously outlined in Zhang et al. [32]. In brief, genomic DNA (1 μg) was digested by EcoRI (New England Biolabs, Ipswich, MA, USA), which recognizes the 5′-GAATTC-3′ sequence. An Illumina P1 adapter containing specific nucleotide barcodes 4–8 bp long was ligated onto the digested DNA. Then, the products from different samples were pooled and randomly fragmented by Covaris E210 (Covaris, Woburn, MA, USA) and selected on an agarose gel of 300–500 bp. The products were purified using a QIAquick PCR Purification Kit. The fragments were end-repaired with an End Repair mix, and then, purified. The repaired DNA was combined with an A-Tailing Mix, then, the Illumina P2 adapters were ligated to the adenylate 3′ ends DNA and followed by purification of the products. Several rounds of PCR amplification with a PCR Primer Cocktail and PCR Master Mix were performed to enrich the adapter-ligated DNA fragments. The PCR products were selected by agarose gel electrophoresis with target fragments and, afterward, purified. The library was qualified using the Agilent Technologies 2100 bioanalyzer and ABI StepOnePlus RealTime PCR System. The qualified libraries were pair-end sequenced on the HiSeq 4000 System (Illumina).

### 4.3. Quality Filtering and SNP Discovery

Raw sequence reads were segregated by barcodes assigned to individuals and low-quality reads, and those that lacked a correct barcode were removed [33]. The reads were first assigned to each individual by the unambiguous barcodes, and the reads without the unique barcodes were discarded. Reads were quality-filtered by removing the adapter sequences and the reads containing greater than 40% low-quality bases (quality value < 20) [34]. All reads were pooled and used for a de novo assembly and SNP calling in ustacks (STACKS v2.55 software pipeline) [35]. We set a minimum stack size of five reads (−m) and a maximum distance between stacks (−M) within a locus as two.

### 4.4. Phylogenetic Analysis

To construct the phylogenetic trees, the genetic distances between the different accessions were calculated based on the high-confidence SNPs extracted from the RAD data. The p-distance, defined as *D_ij_*, between two accessions (*i* and *j*), was calculated using the following equation [36]:(1)$$D_{ij}=\overset{L}{\underset{i=1}{\sum }}d_{ij}^{\left(l\right)}/L$$
where *L* is the length of regions where high-quality SNPs were identified, and *d_ij_* was defined as $d_{ij}^{\left(l\right)}$ = 0 if the genotypes at position *l* for the two accessions were AA and AA, $d_{ij}^{\left(l\right)}$= 0.5 if the genotypes at position *l* were AC and AC (or AA and AC), and $d_{ij}^{\left(l\right)}$ = 1 if the genotypes at position *l* were AA and CC. Neighbor-joining and maximum likelihood phylogenetic trees were constructed by Treebest software, and bootstrap replicates were set to 1000 [28].

The maximum likelihood (ML) tree was constructed using MEGA 5.0 [37] with ultra-fast bootstrap analysis of 1000 replicates. After comparing all the models, JC model was selected as the best-fit substitution model by model selection implemented in MEGA [38].

### 4.5. Principle Component Analysis

Principle component analysis (PCA) was performed using EIGENSOFT based on the SNP dataset [39]. The decomposition of the eigenvectors from the covariance matrix was performed with the R function Eigen, and the significances of the eigenvectors were further investigated with Tracy–Widom tests, using the twstats program in the Eigensoft package.

### 4.6. Genetic Structure Analysis

We identified the population genetic structures using ADMIXTURE 1.3 [40]. We first assumed that the value of ancestral origin is *K* [41]. The difference in the *K* value indicated whether different individuals have the same genetic background and whether the ancestral components are consistent between them. In the analysis of the population genetic structure, each individual was a column of histograms in the figure, and color differences were used to indicate the ancestral composition of different individuals. Individuals with the same color evolved from the same genetic background, and different color ratios of two individuals indicated that the pedigree compositions of the two individuals were different. If a sample always comprised a single color, then, it indicated that there was no hybridization in the sample. Conversely, if a sample comprised multiple colors, then this individual was likely to be a hybrid of several ancestral subpopulations [42]. The individual ancestry proportion was calculated 10,000 times from a given number of inferred populations (*K*) based on the maximum likelihood algorithm. The *K* values were set from two to seven.

ADMIXTURE 1.3 is a calculation method based on the Bayesian model [43]. Cross-validation error will be generated for each *K* value simulation result. The minimum value of cross-validation error corresponds to the optimal *K* value, and the fitting result at this time is closest to the real situation of the population [44].

### 4.7. Ancestral State Reconstruction of Morphological Characters

In this study, we selected two important taxonomic characters of interest, reported in previous studies [2,29], for ancestral state reconstruction over molecular phylogeny. We traced the evolution of two important characters of *Ribes* L.: 1. Gland: (0) absent (1) present; 2. Inflorescence: (0) raceme (1) umbel; (2) short raceme. The character states were optimized onto the tree generated from the RAD-seq dataset in Mesquite v. 2.73 [45] using the maximum likelihood criterion with the Markov k-state one-parameter (Mk1) model [46].

## 5. Conclusions

Our study revealed the genetic relationships of *Ribes* L. species in China by RAD-seq, with a total of 2,451,454 SNPs obtained from 30 species of *Ribes* L. Thus, these 30 species were divided into six branches, i.e., six subgenera, which were not fully consistent with the traditional phenetic sectional division of the genus. Our study was in favor of the separation of subg. *Coreosma* from subg. *Ribesia* and subg. *Hemibotrya* from subg. *Berisia*. Moreover, the results supported subg. *Grossularia* as an independent subgenus. Likewise, the subg. *Grossularia*, subg. *Hemibotrya*, subg. *Coreosma*, and subg. *Berisia* were verified as monophyletic groups by analyzing the genetic relationships and backgrounds of each subgenus. The subg. *Ribesia* had a more complex genetic background, with frequent interspecific hybridizations. *Ribes griffithii*, which originally belonged to subg. *Ribesia*, had a hybrid genetic background. However, the sample size must be expanded for further validation of the rationale to upgrade *R. griffithii* to an independent subgenus.

## Figures and Tables

**Figure 1 plants-12-00829-f001:**
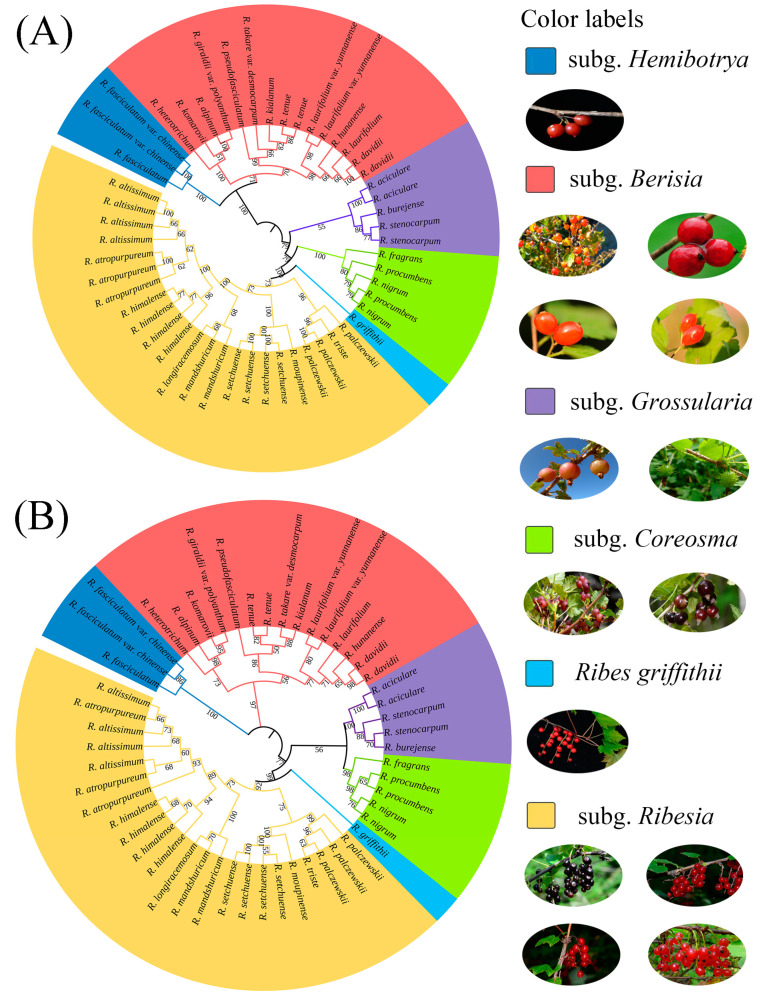
Neighbor-joining phylogenetic tree (**A**) and maximum likelihood phylogenetic tree (**B**) were reconstructed with bootstrap values calculated through 1000 iterations based on 2,451,454 identified SNPs. The numbers on the branches are the related bootstrap supports. Dark blue—subg. *Hemibotrya*, red—subg. *Berisia*, purple—subg. *Grossularia*, green—subg. *Coreosma*, light blue—*Ribes griffithii*, yellow—subg. *Ribesia*.

**Figure 2 plants-12-00829-f002:**
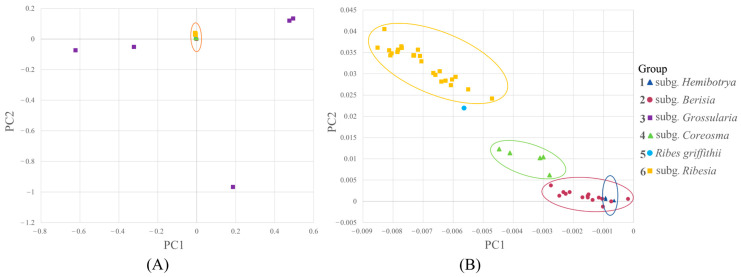
(**A**) Principal component analysis of 30 *Ribes* L. species. (**B**) Principal component analysis of 27 *Ribes* L. species.

**Figure 3 plants-12-00829-f003:**
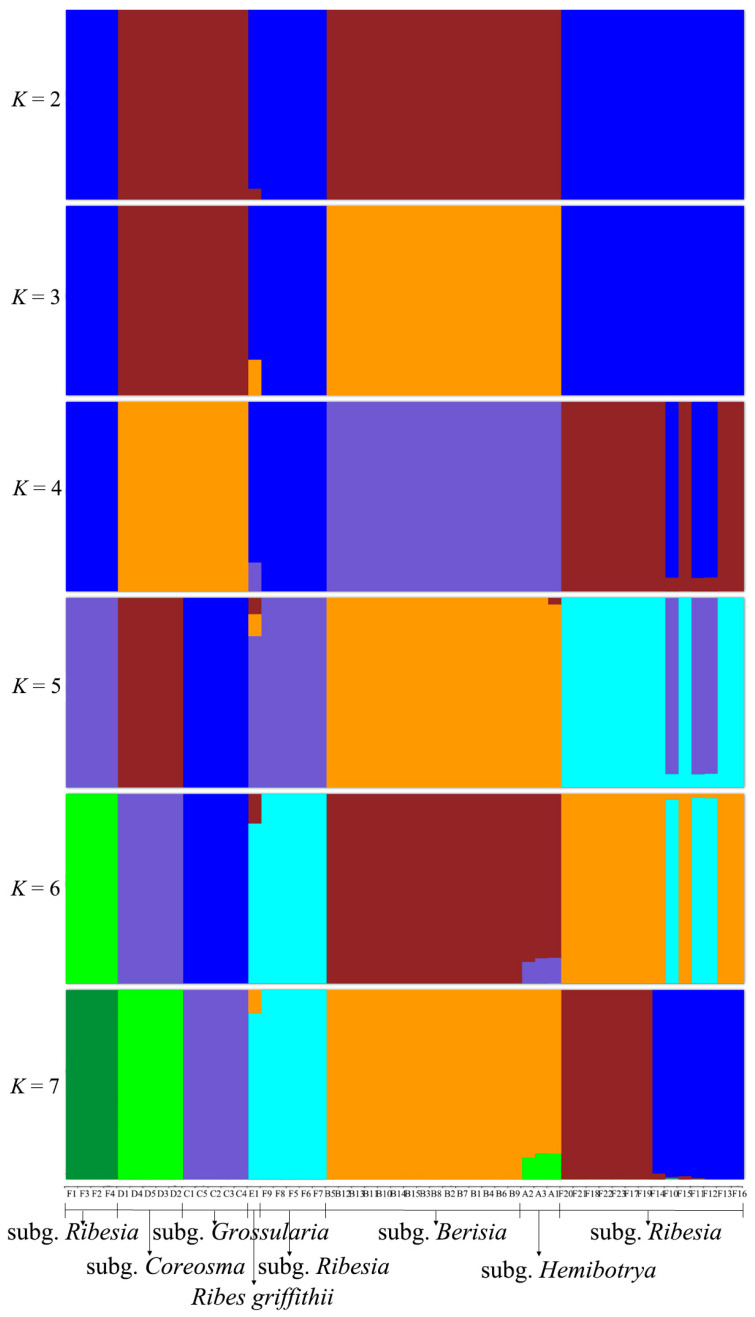
The genetic structure of 52 *Ribes* L. samples. Different ancestral populations are distinguished by different colors. Each individual is a column of histograms in the figure, and the difference in color is used to indicate the ancestral composition of different individuals.

**Figure 4 plants-12-00829-f004:**
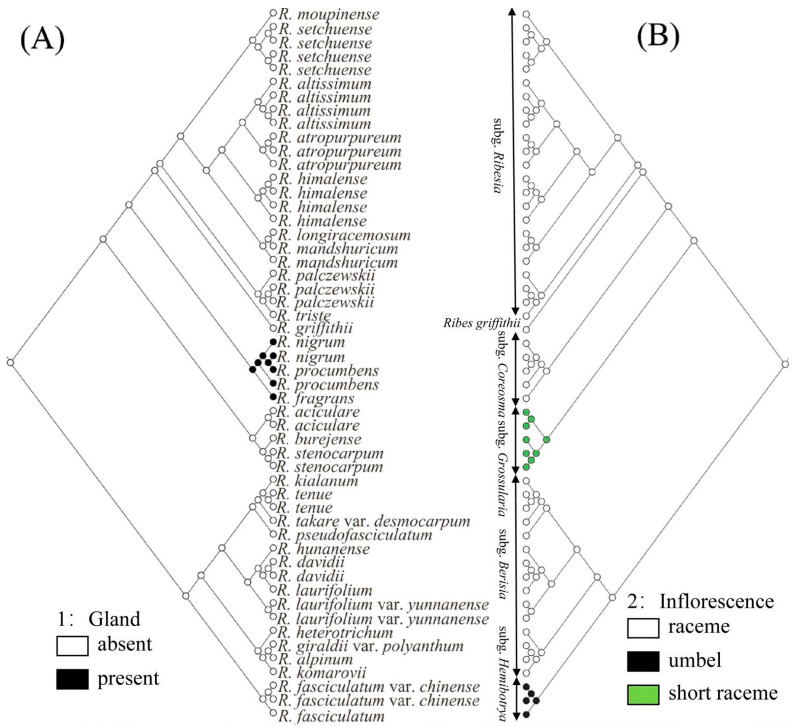
Ancestral state reconstructions for (**A**) gland and (**B**) inflorescence based on the Mk1 model of the neighbor-joining tree. The corresponding color keys identify extant possible ancestral character states. Pie diagrams at internal nodes indicate the relative probabilities for each alternative state.

**Figure 5 plants-12-00829-f005:**
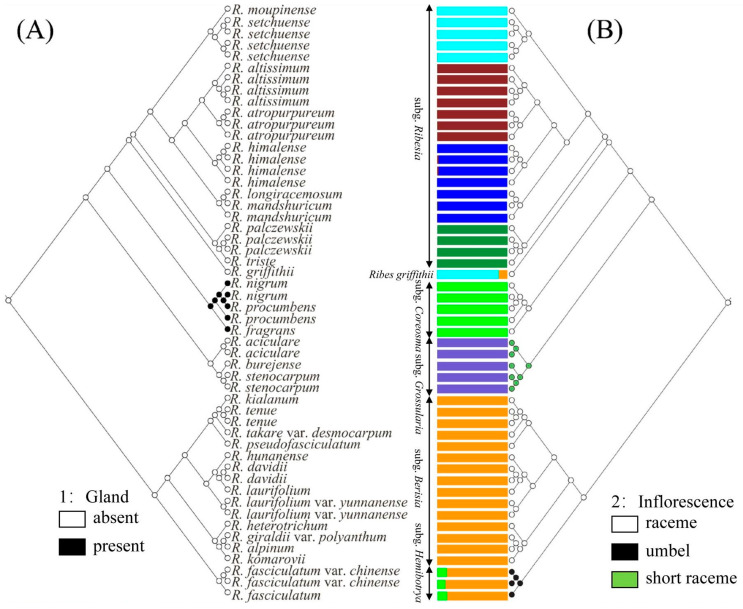
Combination figure of genetic structure, phylogenetic tree, and ancestral morphological reconstruction of 52 samples based on RAD-seq. Reconstruction of ancestral states for (**A**) gland and (**B**) inflorescence using Mesquite. The corresponding colour identify extant possible ancestral character states, which is consistent with Figure 3. Each individual is a column of histograms in the figure, and the difference in colour is used to indicate the ancestral composition of different individuals.

**Figure 6 plants-12-00829-f006:**
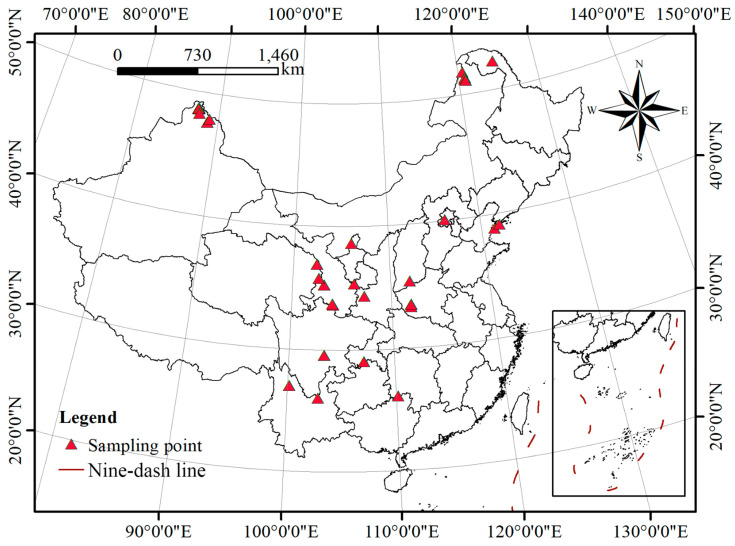
*Ribes* L. sampling sites in China.

**Table 1 plants-12-00829-t001:** The clean reads number, Q_20_, homo SNPs, hete SNPs, and total SNPs for *Ribes* L. samples were measured by RAD-seq and their corresponding code, scientific name.

Code	Scientific Name	Clean Reads Number (M)	Q_20_ (%)	Homo SNPs	Hete SNPs	Total SNPs
A1	*Ribes fasciculatum*	7.68	98.06	18,169	264	18,433
A2	*Ribes fasciculatum* var. *chinense*	10.56	98.08	23,189	322	23,511
A3	*Ribes fasciculatum* var. *chinense*	14.26	98.01	25,609	577	26,186
B1	*Ribes heterotrichum*	6.22	98	30,028	554	30,582
B2	*Ribes komarovii*	17.2	97.89	33,558	797	34,355
B3	*Ribes alpinum*	8.38	97.96	26,143	832	26,975
B4	*Ribes giraldii* var. *polyanthum*	4.32	98.03	25,384	496	25,880
B5	*Ribes pseudofasciculatum*	0.7	97.98	4982	66	50,48
B6	*Ribes takare* var. *desmocarpum*	14.97	97.86	33,613	662	34,275
B7	*Ribes kialanum*	6.95	98.18	29,730	627	30,357
B8	*Ribes tenue*	9.34	97.82	25,164	523	25,687
B9	*Ribes tenue*	12.47	98.1	33,332	843	34,175
B10	*Ribes laurifolium* var. *yunnanense*	9.62	97.89	21,561	313	21,874
B11	*Ribes laurifolium* var. *yunnanense*	10.65	98.02	23,175	368	23,543
B12	*Ribes hunanense*	10.41	98.17	25,959	431	26,390
B13	*Ribes laurifolium*	6.96	98.02	18,367	243	18,610
B14	*Ribes davidii*	7.66	98.01	18,560	301	18,861
B15	*Ribes davidii*	9.76	97.97	23,034	605	23,639
C1	*Ribes aciculare*	10.83	97.84	127,633	14,111	141,744
C2	*Ribes aciculare*	9.97	97.94	126,955	12,425	139,380
C3	*Ribes burejense*	15.79	97.84	142,706	13,211	155,917
C4	*Ribes stenocarpum*	2.07	98.01	141,709	10,065	151,774
C5	*Ribes stenocarpum*	18.36	97.89	144,656	23,607	168,263
D1	*Ribes fragrans*	11.8	98.12	45,922	1578	47,500
D2	*Ribes procumbens*	19.07	98.07	38,844	2554	41,398
D3	*Ribes nigrum*	8.58	98.14	36,776	1009	37,785
D4	*Ribes procumbens*	12.39	98.14	42,188	1394	43,582
D5	*Ribes nigrum*	22.03	97.98	47,442	1601	49,043
E1	*Ribes griffithii*	8.52	98.18	40,367	849	41,216
F1	*Ribes palczewskii*	9.33	98.05	42,553	1036	43,589
F2	*Ribes tenue*	11.72	98.14	43,108	1399	44,507
F3	*Ribes palczewskii*	9.52	98.06	41,924	1114	43,038
F4	*Ribes palczewskii*	21.29	97.94	45,801	1590	47,391
F5	*Ribes moupinense*	7.57	98.16	40,291	805	41,096
F6	*Ribes setchuense*	12.04	97.87	42,788	1120	43,908
F7	*Ribes setchuense*	15.75	97.89	43,072	1100	44,172
F8	*Ribes setchuense*	14.1	97.94	44,005	1365	45,370
F9	*Ribes setchuense*	6.49	97.72	36,222	741	36,963
F10	*Ribes mandshuricum*	14.35	97.85	44,773	1078	45,851
F11	*Ribes mandshuricum*	15.01	98.16	45,074	988	46,062
F12	*Ribes longiracemosum*	9.85	98.07	36,479	631	37,110
F13	*Ribes himalense*	11.21	97.85	35,869	801	36,670
F14	*Ribes himalense*	6.02	98.03	39,633	834	40,467
F15	*Ribes himalense*	15.92	98.07	45,382	1185	46,567
F16	*Ribes himalense*	15.03	98.06	37,404	947	38,351
F17	*Ribes atropurpureum*	12.42	98	43,556	1244	44,800
F18	*Ribes atropurpureum*	13.9	98.06	44,863	1240	46,103
F19	*Ribes atropurpureum*	10.93	98.02	42,549	1254	43,803
F20	*Ribes altissimum*	6.96	98.14	34,220	721	34,941
F21	*Ribes altissimum*	14.78	97.73	42,454	1232	43,686
F22	*Ribes altissimum*	14.17	98.06	44,704	1276	45,980
F23	*Ribes altissimum*	14.48	98.07	43,845	1201	45,046
Average		11.35	98.00	45,314	2399	47,713
Total		590.36		2,335,324	116,130	2,451,454

**Table 2 plants-12-00829-t002:** Code, scientific name, geographic location, altitude, latitude, and longitude for *Ribes* L. samples collected in China.

Code	Scientific Name	Geographic Location	Altitude (Meters)	Latitude	Longitude
A1	*Ribes fasciculatum*	Nanzhao, Henan	613	33°33′27″	111°58′04″
A2	*Ribes fasciculatum* var. *chinense*	Beijing	88	40°04′32″	116°13′58″
A3	*Ribes fasciculatum* var. *chinense*	Dalian, Liaoning	352	38°50′21″	121°26′58″
B1	*Ribes heterotrichum*	Fuhai, Xinjiang	916	47°34′59″	88°44′36″
B2	*Ribes komarovii*	Lishan, Shanxi	2358	35°24′58″	111°59′12″
B3	*Ribes alpinum*	Beijing	88	40°04′32″	116°13′58″
B4	*Ribes giraldii* var. *polyanthum*	Dalian, Liaoning	105	39°06′05″	122°00′03″
B5	*Ribes pseudofasciculatum*	Xining, Qinghai	2332	36°56′37″	102°27′49″
B6	*Ribes takare* var. *desmocarpum*	Zhouqu, Gansu	2578	33°43′50″	104°06′12″
B7	*Ribes kialanum*	Lijiang, Yunnan	3812	27°01′34″	100°10′43″
B8	*Ribes tenue*	Hezheng, Gansu	2684	35°15′14″	103°14′27″
B9	*Ribes tenue*	Xining, Qinghai	2518	35°47′37″	102°40′37″
B10	*Ribes laurifolium* var. *yunnanense*	Leshan, Sichuan	3099	29°34′26″	103°21′23″
B11	*Ribes laurifolium* var. *yunnanense*	Leshan, Sichuan	3099	29°34′26″	103°21′23″
B12	*Ribes hunanense*	Chengbu, Hunan	1669	26°09′39″	110°11′34″
B13	*Ribes laurifolium*	Leshan, Sichuan	3099	29°34′26″	103°21′23″
B14	*Ribes davidii*	Nanchuan, Chongqing	726	29°03′19″	107°08′04″
B15	*Ribes davidii*	Nanchuan, Chongqing	726	29°03′19″	107°08′04″
C1	*Ribes aciculare*	Burqin, Xinjiang	1240	48°12′29″	87°37′10″
C2	*Ribes aciculare*	Fuhai, Xinjiang	916	47°34′59″	88°44′36″
C3	*Ribes burejense*	Nanzhao, Henan	1349	33°32′04″	111°57′28″
C4	*Ribes stenocarpum*	Xining, Qinghai	2332	36°56′37″	102°27′49″
C5	*Ribes stenocarpum*	Guyuan, Ningxia	2928	35°21′51″	106°19′54″
D1	*Ribes fragrans*	Moerdaoga, Inner Mongolia	1256	51°22′43″	120°49′48″
D2	*Ribes procumbens*	Daxing’anling, Heilongjiang	1001	52°20′28″	124°42′12″
D3	*Ribes nigrum*	Hemu, Xinjiang	1389	48°34′27″	87°21′32″
D4	*Ribes procumbens*	Xiniuerhe, Inner Mongolia	707	51°15′51″	120°47′29″
D5	*Ribes nigrum*	Ergun, Inner Mongolia	1404	51°09′19″	120°54′37″
E1	*Ribes griffithii*	Lijiang, Yunnan	3812	27°01′34″	100°10′43″
F1	*Ribes palczewskii*	Hemu, Xinjiang	1099	48°34′13″	87°25′43″
F2	*Ribes tenue*	Moerdaoga, Inner Mongolia	869	51°09′19″	120°54′37″
F3	*Ribes palczewskii*	Xiniuerhe, Inner Mongolia	533	51°52′41″	120°37′56″
F4	*Ribes palczewskii*	Ergun, Inner Mongolia	1404	51°09′19″	120°54′37″
F5	*Ribes moupinense*	Kunming, Yunnan	3419	26°04′46″	102°50′05″
F6	*Ribes setchuense*	Zhouqu, Gansu	2705	33°39′45″	104°09′41″
F7	*Ribes setchuense*	Zhouqu, Gansu	2705	33°39′45″	104°09′41″
F8	*Ribes setchuense*	Hezheng, Gansu	2684	35°15′14″	103°14′27″
F9	*Ribes setchuense*	Hezheng, Gansu	2684	35°15′14″	103°14′27″
F10	*Ribes mandshuricum*	Nanzhao, Henan	1372	33°21′11″	111°57′17″
F11	*Ribes mandshuricum*	Guyuan, Ningxia	2928	35°21′51″	106°19′54″
F12	*Ribes longiracemosum*	Baoji, Shaanxi	568	34°21′22″	107°18′21″
F13	*Ribes himalense*	Hezheng, Gansu	2684	35°15′14″	103°14′27″
F14	*Ribes himalense*	Xining, Qinghai	2332	36°56′37″	102°27′49″
F15	*Ribes himalense*	Xining, Qinghai	2229	35°48′10″	102°40′53″
F16	*Ribes himalense*	Yinchuan, Ningxia	3556	38°37′54″	106°03′35″
F17	*Ribes atropurpureum*	Burqin, Xinjiang	1245	48°37′05″	87°30′03″
F18	*Ribes atropurpureum*	Burqin, Xinjiang	1245	48°37′05″	87°30′03″
F19	*Ribes atropurpureum*	Fuhai, Xinjiang	1489	47°52′05″	88°56′56″
F20	*Ribes altissimum*	Hemu, Xinjiang	1446	48°33′07″	87°28′24″
F21	*Ribes altissimum*	Hemu, Xinjiang	1446	48°33′07″	87°28′24″
F22	*Ribes altissimum*	Hemu, Xinjiang	1099	48°34′13″	87°25′43″
F23	*Ribes altissimum*	Hemu, Xinjiang	1099	48°34′13″	87°25′43″

## Data Availability

Not applicable.

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
