# Peer review of "A Phylogenetic and Morphological Evolution Study of Ribes L. in China Using RAD-Seq"

_plants, 2023, doi:10.3390/plants12040829_

Round 1

Reviewer 1 Report

The study aims to clarify the phylogenetic relationships of Ribes L. taxa in China, and further provide a reference for a revised global classification of it. The phylogenetic analysis of 52 Ribes L. samples from 30 species was constructed based on RAD-seq analysis and two taxonomic characters were selected for ancestral state reconstruction over the molecular phylogeny. 

The study is interesting and several results are innovative. The text is clear and well-structured. However, some doubts about the analysis emerged during the reading.

The authors have chosen to set the K value in Genetic structure analysis from 2 to 7. However, I don’t understand How was selected the best K. I suggest using a method, based on the likelihood, to obtain the best K value (e.g. Evanno method or others). Several papers discuss How does one select the suitable K value for determining the genetic cluster and this issue should be clarified in the manuscript.

The authors choose to produce a Phylogenetic tree (Neighbor-joining ) using the p-distance. The authors should use an ML method using several runs and then the trees obtained (NJ and ML best) should be compared. A scarce exploration of likelihood could don’t allow to find the best Tree. I suggest using RAXML or similar software.

Figure 2 is not clear. For example, the position of imagines of Ribes on the clades is not clear. The authors can use the lines to delimit the subgens. The bootstrap is not visible on the nodes and the circles are all similar. I suggest inserting the number of bootstrap  on the nodes or to use other symbols. Moreover, the authors should to use the same colours of Ribes groups for all figures. This could help the readers.

When you have defined the K number for the cluster analyses, the authors should to combine the result of ADMIXTURE with the Phylogenetic tree and with morphological/taxonomic results in a new figure at the end of the Discussion. In this way, the new figure obtained should help to understand the Discussion.  

I am not convinced that this study will provide important information on the hybridization history. The authors should use specific analyses to define the level of hybridization that in this version I don't find. I suggest to clarify this issue using specific analysis or to delete these sentences from the manuscript.

Author Response

Dear Professor,

    Thank you very much for your careful review and for giving us constructive comments and suggestions of our manuscript ID: plants-2203534, entitled "A phylogenetic and morphological evolution study of Ribes L. in China based on RAD-seq", which have helped us in depth to improve the quality of our paper. Now, we have revised the manuscript carefully according to the comments and re-submitted it.

   The point-to-point responses to the comments are as the following attached file showed and the revised parts under your guidance were all highlighted in blue in our revised manuscript.

Reviewer 2 Report

The manuscript is well structured and the subject matter very interesting. 

You will find all comments in the attached file. 

Some of the methods used do not have references, they absolutely must be added. 

the tables are not always well explained.

I have a doubt/curiosity, why if there are 59 species and 30 varieties of Ribes in China were not all analysed but only 30? are they part of a different subgenus or were no samples found for examination? write in materials and methods why, justify the choice.

in general it is a good manuscript with some gaps to be filled before publication.

Author Response

Dear Professor,

    Thank you very much for your careful review and for giving us constructive comments and suggestions of our manuscript ID: plants-2203534, entitled "A phylogenetic and morphological evolution study of Ribes L. in China based on RAD-seq", which have helped us in depth to improve the quality of our paper. Now, we have revised the manuscript carefully according to the comments and re-submitted it.

   The point-to-point responses to the comments are as the following attached file showed and the revised parts under your guidance were all highlighted in yellow in our revised manuscript.

Round 2

Reviewer 1 Report

Accept in present form

Reviewer 2 Report

Dear authors, 

I have read the manuscript with the revisions made, thank you for answering all my questions.

I consider the manuscript ready for publication.

You have done an excellent job, congratulations.